# Inter-membrane association of the Sec and BAM translocons for bacterial outer-membrane biogenesis

Sara Alvira[1†], Daniel W Watkins[1†], Luca A Troman[1], William J Allen[1], James S Lorriman[1], Gianluca Degliesposti[2], Eli J Cohen[3], Morgan Beeby[3], Bertram Daum[4,5], Vicki AM Gold[4,5], J Mark Skehel[2], Ian Collinson[1]*

[1]School of Biochemistry, University of Bristol, Bristol, United Kingdom; [2]Biological Mass Spectrometry and Proteomics, MRC Laboratory of Molecular Biology, Cambridge, United Kingdom; [3]Department of Life Sciences, Imperial College London, London, United Kingdom; [4]Living Systems Institute, University of Exeter, Exeter, United Kingdom; [5]College of Life and Environmental Sciences, University of Exeter, Exeter, United Kingdom

**Abstract** The outer-membrane of Gram-negative bacteria is critical for surface adhesion, pathogenicity, antibiotic resistance and survival. The major constituent – hydrophobic β-barrel *Outer-Membrane Proteins* (OMPs) – are first secreted across the inner-membrane through the Sec-translocon for delivery to periplasmic chaperones, for example SurA, which prevent aggregation. OMPs are then offloaded to the β-*Barrel Assembly Machinery* (BAM) in the outer-membrane for insertion and folding. We show the *Holo-TransLocon* (HTL) – an assembly of the protein-channel core-complex SecYEG, the ancillary sub-complex SecDF, and the membrane 'insertase' YidC – contacts BAM through periplasmic domains of SecDF and YidC, ensuring efficient OMP maturation. Furthermore, the proton-motive force (PMF) across the inner-membrane acts at distinct stages of protein secretion: (1) SecA-driven translocation through SecYEG and (2) communication of conformational changes via SecDF across the periplasm to BAM. The latter presumably drives efficient passage of OMPs. These interactions provide insights of inter-membrane organisation and communication, the importance of which is becoming increasingly apparent.

*For correspondence:
ian.collinson@bristol.ac.uk

†These authors contributed equally to this work

Competing interests: The authors declare that no competing interests exist.

## Introduction

Outer-membrane biogenesis in Gram-negative bacteria (reviewed in *Konovalova et al., 2017*) requires substantial quantities of protein to be exported, a process which begins by transport across the inner plasma membrane. Precursors of β-barrel *Outer-Membrane Proteins* (OMPs) with cleavable N-terminal signal-sequences are targeted to the ubiquitous Sec-machinery and driven into the periplasm by the ATPase SecA and the trans-membrane proton-motive force (PMF) (*Brundage et al., 1990*; *Collinson, 2019*; *Lill et al., 1989*; *Cranford-Smith and Huber, 2018*). Upon completion, the pre-protein signal-sequence is proteolytically cleaved (*Josefsson and Randall, 1981*; *Chang et al., 1978*), releasing the mature unfolded protein into the periplasm. The emergent protein is then picked up by periplasmic chaperones, such as SurA and Skp, which prevent aggregation (*McMorran et al., 2013*; *Sklar et al., 2007*), and somehow facilitate delivery to the β-*Barrel Assembly Machinery* (BAM) for outer-membrane insertion and folding (*Voulhoux et al., 2003*; *Wu et al., 2005*).

In *Escherichia coli*, BAM consists of a membrane protein complex of subunits BamA-E, of known structure (*Bakelar et al., 2016*; *Gu et al., 2016*; *Iadanza et al., 2016*). The core component, BamA, is a 16 stranded β-barrel integral membrane protein, which projects a large periplasmic stretch of 5

*PO*lypeptide *TR*anslocation-Associated (POTRA) domains into the periplasm. BamB-E are peripheral membrane lipoproteins anchored to the inner leaflet of the OM. In spite of the structural insights, the mechanism for BAM-facilitated OMP insertion is unknown (*Ricci and Silhavy, 2019*).

The bacterial periplasm is a challenging environment for unfolded proteins, so complexes spanning both membranes are critical for efficient delivery through many specialised secretion systems (*Green and Mecsas, 2016*). How do enormous quantities of proteins entering the periplasm via the general secretory pathway (Sec) efficiently find their way through the cell envelope to the outer-membrane? From where is the energy derived to facilitate these trafficking processes some distance from the energy transducing inner-membrane, and in an environment lacking ATP? Could it be achieved by a direct interaction between chaperones, and the translocons of the inner (Sec) and outer (BAM) membranes?

The core-translocon, SecYEG, does not possess periplasmic domains of sufficient size to mediate such an interaction (*Van den Berg et al., 2004*). However, the Holo-TransLocon (HTL) contains the ancillary sub-complex SecDF and the membrane protein 'insertase' YidC (*Duong and Wickner, 1997*; *Schulze et al., 2014*), both of which contain periplasmic extensions potentially large enough to reach the POTRA domains of BamA.

SecDF is a member of the so-called root nodulation division (RND) superfamily of PMF-driven transporters (reviewed in *Tseng et al., 1999*). It is a highly conserved component of the bacterial Sec translocon, wherein it has long been known to facilitate protein secretion (*Duong and Wickner, 1997*; *Economou et al., 1995*; *Pogliano and Beckwith, 1994*). While fellow component of the HTL – YidC – is essential for membrane protein insertion, and thus indispensable (*Samuelson et al., 2000*; *Scotti et al., 2000*), mutants of *secD* and *secF* are not fatal but severely compromised and cold-sensitive (*Gardel et al., 1987*), presumably due to deficiencies in envelope biogenesis. The cause of this has been ascribed to a defect in protein transport across the inner membrane.

In keeping with other members of the RND family, like AcrB (*Eicher et al., 2014*), SecDF confers PMF stimulation of protein secretion (*Arkowitz and Wickner, 1994*). Different structures of SecDF show the large periplasmic domains in different conformational states (*Furukawa et al., 2017*; *Mio et al., 2014*; *Tsukazaki et al., 2011*), affected by altering a key residue of the proton transport pathway (SecD$_{D519N}$ – *E. coli* numbering) (*Furukawa et al., 2017*). On this basis, an elaborate mechanism has been proposed whereby PMF-driven conformational changes, at the outer surface of the inner-membrane, pick up and pull polypeptides as they emerge from the protein-channel exit site of SecY. Yet, ATP- and PMF-driven translocation across the inner-membrane does not require SecDF or YidC; SecYEG and SecA will suffice (*Brundage et al., 1990*; *Schulze et al., 2014*). Evidently then, there must be two PMF-dependent components of protein secretion: an early stage dependent only on SecYEG/SecA and another later event regulated by an AcrB-like SecDF activity. This distinction has not been fully appreciated.

This study explores the role of the ancillary components of the Sec machinery for protein secretion, and for downstream trafficking through the periplasm for delivery to the outer-membrane and OMP maturation. In particular, we examine the possibility of a direct interaction between the HTL and BAM machineries to facilitate protein transport through the envelope. The basic properties and structure of the inter-membrane super-complex are investigated, as well as its importance for OMP folding and insertion. The implications of this interaction and its modulation caused by proton transport through SecDF are profound. Thus, we consider their consequences for the mechanism of protein transport through the Sec and BAM machineries, and for outer-membrane biogenesis.

## Results

### Co-fractionation and immunoprecipitation highlight an interaction between the Sec and BAM machineries

Total *E. coli* membranes from cells over-producing either SecYEG or HTL were prepared and fractionated by sucrose gradient centrifugation to separate the inner- and outer-membranes (*Figure 1a*). We first sought to determine the precise locations of the respective inner- and outer-membrane proteins in the fractions; SDS-PAGE analysis and staining for total protein revealed the presence of SecY in the lighter inner-membrane fractions (*Figure 1—figure supplement 1a*, yellow asterisk – left panel). Heating the fractions (required to unfold outer-membrane proteins) prior to

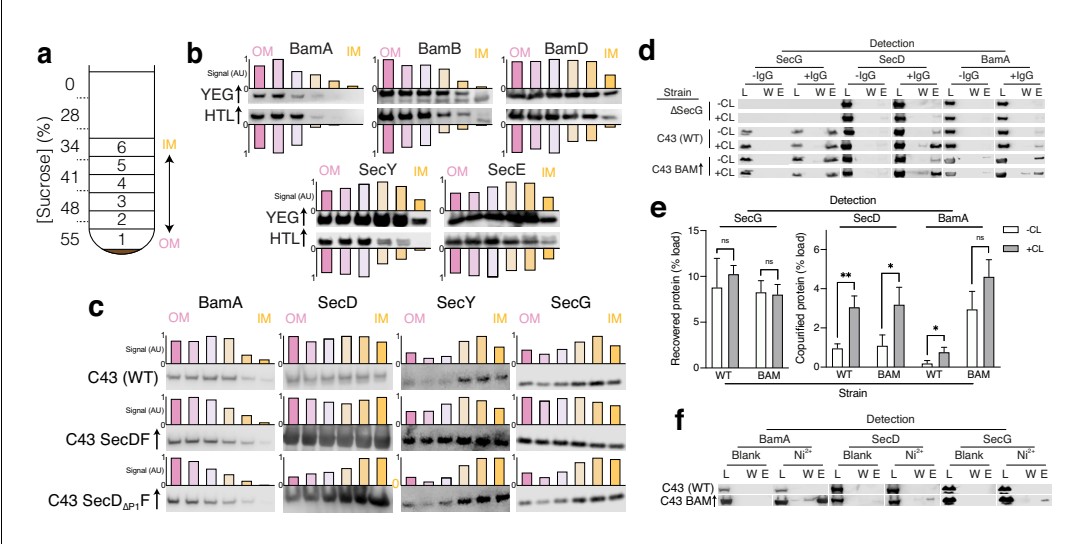

**Figure 1.** Identification of interactions between HTL and BAM. (**a**) Schematic representation of sucrose gradient centrifugation tube for fractionation of *E. coli* total membranes. Numbers 1–6 indicate the fractions taken for SDS-PAGE and immunoblotting shown in (**b, c** and **d**). (**b, c**) Immunoblots of fractions produced as shown in **a** for membranes of (**b**): *E. coli* C43 overproducing either SecYEG or HTL or for (**c**): *E. coli* C43 with no over-expression, and those over-producing either SecDF or SecD$_{\Delta P1}$F (lacking the periplasmic domain 1 (P1) of SecD). To help visualise migration shifts, blotting signal was used to quantify relative abundances of proteins of interest in fractions, shown above or below blots as normalised bar charts, where bars from left (pink) to right (yellow) indicate fractions 1 (OM) – 6 (IM), respectively. (**d**) Co-immunoprecipitations (co-IP) of SecG, SecD, and BamA – pulling with the SecG antibody. Pull-downs were conducted with solubilised crude membrane extracts from *E. coli* C43 (WT), a strain lacking SecG (Δ*secG*), and C43 over-producing BAM. Experiments were conducted in the presence (+CL) and absence (-CL) of cardiolipin. L = load (1% total material), W = final wash before elution (to demonstrate complete washing of affinity resin, 17% of total material) and E = elution (17% of total material). (**e**) Quantification of IPs shown in (**d**). Error bars represent SEM. An unpaired T-test was used to compare samples (p=0.05, n = 3, * = <0.05, ** = <0.01, p values from left to right are 0.4874, 0.8083, 0.0041, 0.0249, 0.0241, and 0.0839). Quantification was performed for cells of *E. coli* C43 (WT) and the same but overproducing BAM (BAM). (**f**) Affinity pull-down of recombinant BamA-His$_6$, SecD, and SecG by nickel chelation all in the presence of cardiolipin. L, W, and E as described in (**d**).

The online version of this article includes the following figure supplement(s) for figure 1:

**Figure supplement 1.** Raw western blots of co-immunoprecipitations and affinity pull-downs.

SDS-PAGE helped reveal the location of the most highly expressed outer-membrane residents (OmpC and OmpF; *Figure 1—figure supplement 1a*, yellow asterisk – right panel). Thus, in these gradients fractions 1–2 mostly contain outer-membranes, and fractions 4–5 are composed mainly of inner-membranes.

Immunoblotting confirmed the presence of the BAM components (BamA, BamB, and BamD), as expected, in outer-membrane fractions (OM; *Figure 1b*). Likewise, the over-produced SecY and SecE subunits mark the fractions containing the core-complex (SecYEG) in the inner-membrane fractions (IM; *Figure 1b*, YEG↑). However, when over-produced as part of HTL, there is a marked shift of their migration peak towards the outer-membrane containing fractions (*Figure 1b*, HTL↑). Interestingly, the over-production of SecDF alone results in a similar effect (*Figure 1c*), where SecD, SecY, and SecG all migrate into the outer-membrane containing fractions. An effect which was lost in comparable experiments where the periplasmic domain of SecD (P1) had been removed (*Figure 1c*). Our interpretation of these experiments is that an interaction between the Sec and Bam complexes, requiring at least the periplasmic domains of SecD (and most likely SecF and YidC), causes an association of inner- and outer-membrane vesicles reflected in the shift we observe.

To further examine this interaction, we extracted native membranes with a mild detergent for immuno-precipitation (IP) using a monoclonal antibody raised against SecG. The pull-downs were then probed for native interacting partners by western blotting (*Figure 1d,e*; *Figure 1—figure supplement 1b*). As expected, SecG (positive control) and SecD of HTL co-immuno-precipitated. Crucially, BamA could also be detected. The specificity of the association was demonstrated by controls omitting the SecG antibody or the SecG protein (produced from membranes extracts of a Δ*secG*

strain; *Nishiyama et al., 1994*), wherein non-specific binding was either undetectable, or considerably lower than the specific co-immuno-precipitant (*Figure 1d*). When BAM was over-produced the yield of BamA recovered in the IPs increased accordingly (*Figure 1d,e*; *Figure 1—figure supplement 1b*).

In a similar experiment, a hexa-histidine-tagged BamA was used to isolate BAM from cells over-producing the complex. Western blots showed that BamA co-purified, as expected, with additional components of the BAM complex (BamB and BamD), and crucially also with SecD and SecG of the HTL (*Figure 1f*; *Figure 1—figure supplement 1c*). Again, controls (omitting $Ni^{2+}$, or recombinant $His_6$-BamA) were reassuringly negative.

## Interaction between HTL and BAM is cardiolipin dependent

The phospholipid cardiolipin (CL) is known to be intimately associated with energy-transducing systems, including the Sec-machinery, for both complex stabilisation and efficient transport (*Schulze et al., 2014*; *Corey et al., 2018*; *Gold et al., 2010*). For this reason, the IP experiments above were augmented with CL. On omission of CL, the interactions of SecG with SecD and BamA were reduced approximately three- and fivefold, respectively (*Figure 1d,e*; *Figure 1—figure supplement 1b*). This lipid-mediated enhancement of the SecG-SecD interaction is consistent with our previous finding that CL stabilises HTL (*Schulze et al., 2014*) and shows it also holds true for the HTL-BAM interaction. *Apropos*, CL has been shown to be associated with the BAM complex (*Chorev et al., 2018*).

## HTL and BAM interact to form an assembly large enough to bridge the inner- and outer-membranes

To confirm the interaction between the Sec and BAM machineries, the purified complexes were subjected to glycerol gradient centrifugation. When mixed together, HTL and BAM co-migrated towards higher glycerol concentrations, beyond those attained by the individual complexes (*Figure 2a*, yellow asterisk) and consistent with the formation of a larger complex due to an interaction between the two. The interaction is clear but not very strong, wherein only a fraction of the HTL and BAM associates. This low affinity is likely due to the required transient nature of the association between the two translocons *in vivo*, and also because of the complete breakdown of the inner- and outer-membranes by detergent – required for this experiment. When the experiment was repeated with the individual constituents of HTL: SecDF and YidC, but not SecYEG, were also shown to interact with BAM (*Figure 2—figure supplement 1a–c*, yellow asterisks). Again, the incomplete association suggest their affinity for one another is not high.

Visualisation of the heavy fractions containing interacting HTL and BAM by negative stain electron microscopy (EM) revealed a heterogeneous mixture of small and very large complexes (*Figure 2—figure supplement 2a*, large complexes marked with white arrows). As noted above, this mixed population is probably due to the expected transient nature of the interaction between the two complexes, and/or due to super-complex instability caused by loss of the bilayer and specifically bound phospholipids, for example CL, during purification (see above and below). Even though we augment the material with CL, it is unlikely the full complement of lipids found in the native membrane-bound state are restored.

To overcome this heterogeneity, we stabilised the complex by cross-linking, using GraFix (*Kastner et al., 2008*; *Figure 2—figure supplement 3a*, left). Note that successful stabilisation of the assembly by cross-linking was also demonstrated by size exclusion chromatography – performed for sample preparation for cross-linked mass spectrometry (XL-MS) and cryo-EM (see next section). We confirmed the presence of BAM and HTL constituents in the cross-linked fraction by mass spectrometry (*Figure 2—figure supplement 3a*, right, *Figure 2—figure supplement 3—source data 1*) and subsequently analysed it by negative stain EM, which revealed a marked reduction in the number of dissociated complexes (*Figure 2—figure supplement 2b*). As expected, omitting CL from the preparation results in dissociation of the majority of the large complexes, even with GraFix (*Figure 2—figure supplement 2c*), supporting the above findings regarding CL dependence of the interaction (*Figure 1e*).

The subsequent single-particle analysis of the cross-linked material (*Figure 2—figure supplement 3a*, left, black asterisk; *Figure 2—source data 1*) revealed a remarkable structure large enough

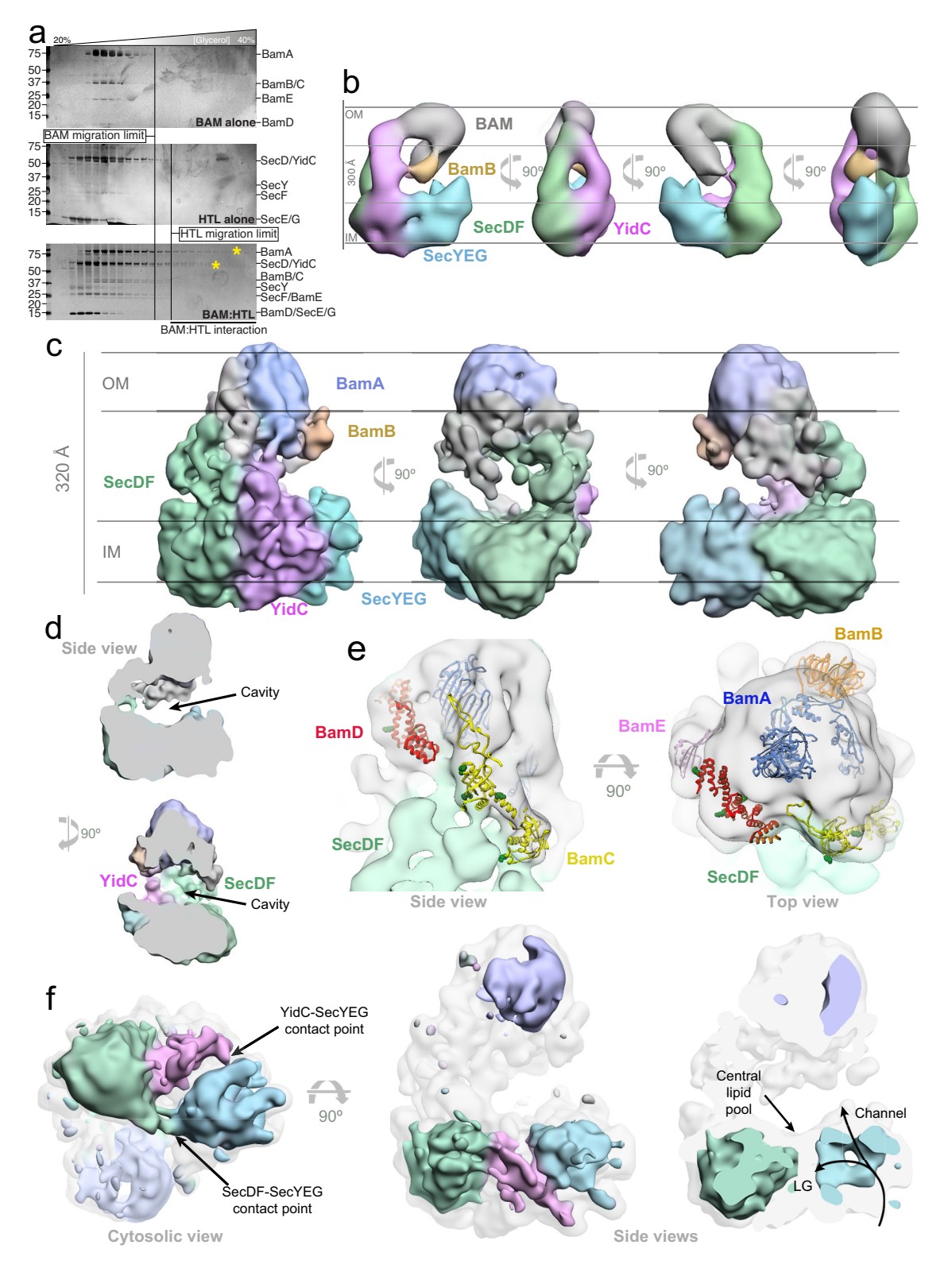

**Figure 2.** 3D characterisation of HTL-BAM by negative stain-EM and cryo-EM in detergent solution, and XL-MS analysis. (a) Silver-stained SDS-PAGE gels of fractions from glycerol centrifugation gradients, with increasingly large complexes appearing in fractions of higher percentage glycerol, (from left to right). Gels of BAM alone (top) HTL alone (middle) and HTL mixed with BAM (bottom) are shown. The fractions of furthest migration of the individual components, as determined in the top two gels, are marked by vertical lines. HTL-BAM components in heavy fractions are marked with a

*Figure 2 continued on next page*

*Figure 2 continued*

yellow asterisk. (**b**) Negative stain analysis of the HTL-BAM complex (37.2 Å resolution) in four representative orthogonal views, with the orientation with respect to the inner and outer membranes inferred. BAM (grey), BamB (orange), SecYEG (cyan), YidC (pink), and SecDF (green) are shown. (**c**) Three orthogonal views of the cryo-EM HTL-BAM complex 3D reconstruction (18 Å resolution). Colours are as in (**a**), but with BamA in blue. (**d**) Side views of the cryo-EM HTL-BAM complex showing the large cavity between the inner-membranes and outer-membrane complexes. (**e**) Close-up of the outer-membrane region of the HTL-BAM complex. The cryo-EM structure (transparent surface) with BamABCDE atomic structures docked (pdb: 5d0q). The position of BamB (orange) was determined directly by negative stain-EM (*Figure 2—figure supplement 3e*). BamA (blue), BamC (yellow), BamD (red), and BamE (pink) are docked according to the HTL-BAM cryo-EM density and XL-MS data (*Figure 2—figure supplement 5c*). Green sphere atoms in BamC and BamD show interacting points with SecD identified by mass apectrometry. (**f**) Lower threshold map of HTL-BAM overlaid with the standard threshold (transparent grey), with the main components coloured as in (**a**). The lateral gate (LG) into the membrane and protein-channel both through SecY and the central lipid pool of the HTL are highlighted.

The online version of this article includes the following source data and figure supplement(s) for figure 2:

**Source data 1.** Parameters of EM analysis of HTL and HTL-BAM structures.
**Figure supplement 1.** Glycerol centrifugation gradients of HTL and BAM components.
**Figure supplement 2.** Negative-stained EM micrographs of the HTL-BAM complex.
**Figure supplement 3.** 3D characterisation and subunit assignment of HTL-BAM by negative stain-EM in detergent solution.
**Figure supplement 3—source data 1.** Mass spectrometry (MS) analysis of the GraFix fractions for image processing.
**Figure supplement 4.** Image processing and classification strategy for the cryo-EM data of the HTL-BAM complex.
**Figure supplement 5.** Sample preparation and XL-MS analysis of HTL-BAM.

(~300×250x150 Å) to contain both Sec and BAM machineries (*Iadanza et al., 2016*; *Botte et al., 2016*), and with a height sufficient to straddle the space between the two membranes (*Figure 2b*; *Figure 2—figure supplement 3b*), especially when considering the plasticity of the periplasm (*Zuber et al., 2008*). Indeed, regions of SecDF and the POTRA domains of BamA have been shown to extend ~60 Å (*Furukawa et al., 2017*) and ~110 Å (*Ma et al., 2019*) respectively, sufficient to bridge this gap.

To assign the locations and orientations of the individual constituents of HTL and BAM, we compared the 3D reconstructions of different sub-complexes: BAM bound to SecYEG-DF (without YidC) (*Figure 2—figure supplement 3c*) or SecDF alone (*Figure 2—figure supplement 3d*). The difference analysis revealed the locations of YidC (*Figure 2b*, pink; *Figure 2—figure supplement 3c*, pink arrow), SecDF (*Figure 2b*, green; *Figure 2—figure supplement 3d*, green arrow), and SecYEG (*Figure 2b*, blue; *Figure 2—figure supplement 3d*, blue arrow) at the bottom of the assembly (assigned as the inner-membrane region). The orientation of BAM relative to SecDF is different in SecDF-BAM compared to HTL-BAM (*Figure 2—figure supplement 3d*, red arrows), possibly due to its known ability to move (see below), and/or the absence of stabilising interactions with the missing HTL components (SecYEG and YidC).

Removing BamB from the complex results in the loss of significant mass in the area designated as the outer-membrane region (*Figure 2b*, orange; *Figure 2—figure supplement 3e*, orange arrow). This confirmed the orientation of the respective inner- and outer-membrane-associated regions and the assignment of the BAM complex as shown in *Figure 2b*. Interestingly, the complex lacking BamB shows a diminishment of the density assigned as YidC (*Figure 2—figure supplement 3e*, pink arrow), suggestive of a mutual interaction between the two.

## Periplasmic domains of the Sec and BAM translocons associate to form a large cavity between the bacterial inner- and outer-membranes

Despite heterogeneity in the sample, we were able to isolate a cross-linked HTL-BAM complex by size exclusion chromatography and produce a low-resolution cryo-EM structure (*Figure 2c*; *Figure 2—figure supplement 4*; *Figure 2—figure supplement 5a* (a similar fraction was used to that marked by the black asterisk) and *Figure 2—source data 1*) with an overall resolution of 18.2 Å. Taken together with the difference map generated by negative stain-EM (*Figure 2b*; *Figure 2—figure supplement 3*), the structure reveals the basic architecture of the assembly and the arrangement of constituent subunits.

The complexity of the image processing resulted in an insufficient number of particles of a single class to attain high resolution. Many factors contribute to this problem: the dynamism of the complex due to the limited contact surface between the HTL and BAM; its inherent mobility necessary

for function; the presence of detergent surrounding the trans-membrane regions of the HTL and BAM components accounting for most of the surface of the assembly; and finally, the absence of inner- and outer-membrane scaffolds. The loss of the fixed double membrane architecture was particularly problematic; during image processing, we found different sub-populations where BAM pivots away from its raised position towards where the inner-membrane would otherwise have been. Obviously this would not happen if restrained by the outer-membrane.

Due to the limited resolution, we deployed cross-linking mass spectroscopy (XL-MS) to verify the contacts between HTL and BAM responsible for inter-membrane contact points. The HTL and BAM complexes were mixed together in equimolar quantities and cross-linked with the lysine-specific reagent DSBU. The reaction mixture was then fractionated by gel filtration chromatography and analysed by SDS-PAGE. A single band corresponding to the cross-linked HTL-BAM complex was detected (*Figure 2—figure supplement 5a*, lower band, black asterisk); note that the isolation of the intact HTL-BAM complex by gel filtration chromatography provides further evidence of a genuine interaction between inner- and outer-membrane translocons. The fractions containing the cross-linked complex were combined and digested prior to LC-MS/MS analysis.

The analysis of mass spectrometry data enabled the detection and mapping of the inter- and intra-molecular protein cross-links within the assembly. The results show an intricate network of interactions, most of which are consistent with the cryo-EM structure, particularly at one side of the assembly between SecD and BamBCD and on the other side between YidC and BamABCD (*Figure 2—figure supplement 5b,c*).

All the constituent proteins of HTL were cross-linked to BAM subunits with the exception of SecG and YajC. Thus, the co-immunoprecipitation and affinity pull-down of SecG together with BamA (described above; *Figure 1d–f*) must have been the result of an indirect interaction, presumably bridged via SecDF-YidC, which interacts with both SecG and BAM. This is consistent with the lack of an interaction of SecYEG alone with the BAM complex (*Figure 2—figure supplement 1a*), and the assignment of the electron microscopy structures (*Figure 2b,c*) – also showing no direct connection between SecYEG and BAM. In this respect, it is interesting to note in the structure that the periplasmic domains of SecD, YidC, and to a lesser degree SecF, extend to establish multiple interactions with the BAM lipoproteins suggesting a pivotal role for these subunits in the formation of the HTL-BAM complex (*Figure 2c,d*). This bridge between the two complexes also helps to define a very large cavity between the inner- and outer-membrane regions (*Figure 2d*).

The BAM complex is recognisable in the cryo-EM structure at the outer-membrane with the expected extensive periplasmic protrusions (*Bakelar et al., 2016*; *Gu et al., 2016*). Some components of the BAM complex, such as BamB, can be unambiguoulsy docked into the cryo-EM structure (*Figure 2c*), localised by negative stain difference mapping (*Figure 2b* and *Figure 2—figure supplement 3e*), and its recognisable β-propeller shape (*Bakelar et al., 2016*; *Gu et al., 2016*). We also suggest the locations of BamA, BamC, and BamD according to the cryo-EM density and the constraints of the XL-MS data (*Figure 2e*; *Figure 2—figure supplement 5c*).

The inner-membrane region of the HTL – while bound to BAM – is much more open than the previous structure of the isolated version (*Botte et al., 2016*). In the new open structure, the locations of the core-complex SecYEG, SecDF, and YidC can be easily distinguished, in which the former two are connected within the membrane by two bridges (*Figure 2f*, left). These connections could be the binding sites of CL and the central lipid pool identified previously, required for structural stability and translocon activity (*Schulze et al., 2014*; *Corey et al., 2018*; *Martin et al., 2019*). Within SecYEG, the protein-channel can be visualised through the centre, along with the lateral gate (LG, required for signal sequence binding and inner-membrane protein insertion) facing towards SecDF, YidC, and the putative central lipid pool (*Martin et al., 2019*; *Figure 2f*, right).

## Cardiolipin, required for super-complex formation, stabilises an 'open' form of the HTL

As mentioned above, the HTL bound to BAM in our EM structure (*Figure 3*, structure ii) seems to be more open when compared to the previously published low-resolution cryo-EM structure (*Botte et al., 2016*) (emd3056; *Figure 3*, structure i) and also displays a more prominent periplasmic region. Preparations of HTL alone, made in this study, contain both a 'compact' state (*Figure 3*, structure iii) similar to that of the previously published structure (*Figure 3*, structure i), as well as a proportion of an 'open' state, with proud periplasmic domains, not previously described

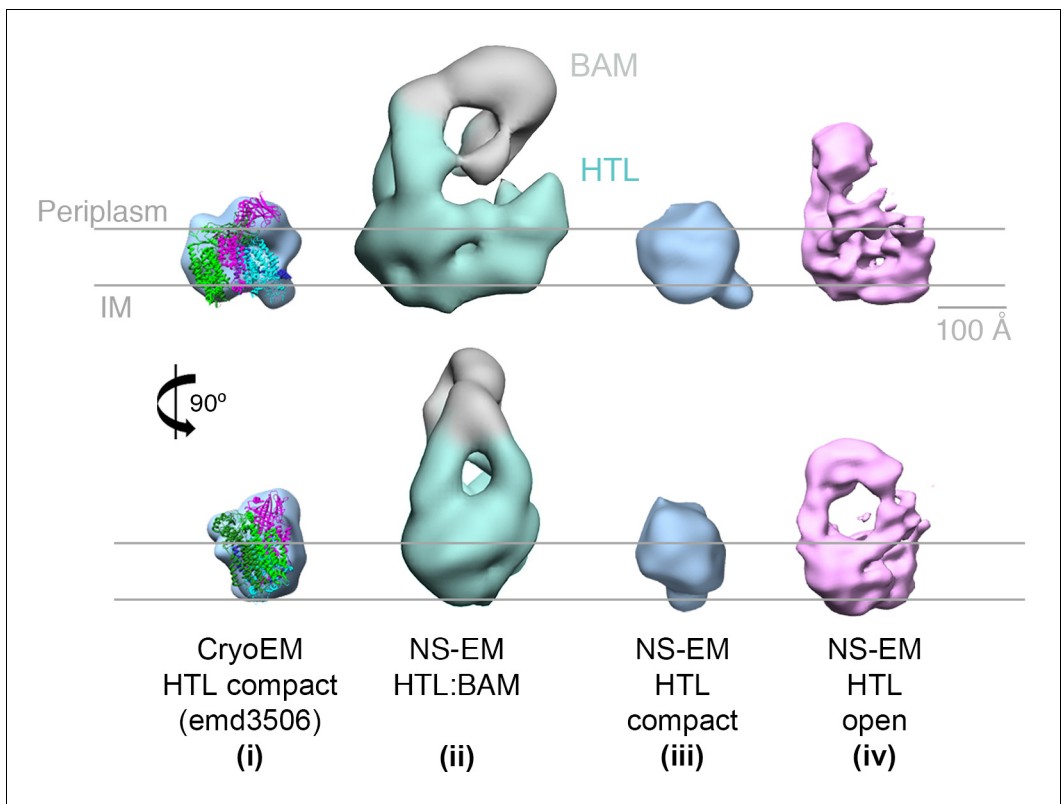

**Figure 3.** EM structures of HTL in 'compact' and 'open' states. Structure and docking of a previously published cryo-EM structure of HTL in the compact state (i) (*Botte et al., 2016*), the HTL-BAM complex (ii), HTL in the 'compact' state (iii) and HTL in the 'open' state (iv); structures (ii - iv) are from this study.
The online version of this article includes the following figure supplement(s) for figure 3:

**Figure supplement 1.** EM field of wild type HTL in different conditions.

(*Figure 3*, structure iv) and apparently more similar to that seen in the HTL-BAM structure (*Figure 3*, structure ii).

The HTL sample used here is extremely pure, of known subunit composition and not prone to oligomerisation (*Schulze et al., 2014*). So, we can rule out that this larger form, assigned as an 'open' state, of HTL is not due to the presence of contaminants, unknown additional partner proteins, or dimerisation. Lipid content within the HTL is critical for proper structure and function, and CL is particularly important for protein translocation through the Sec machinery (*Schulze et al., 2014*; *Corey et al., 2018*; *Gold et al., 2010*; *Hendrick and Wickner, 1991*). Depletion of these core lipids, for instance by detergent extraction, might be expected to cause a collapse of the complex. Therefore, the reason for the presence of these different populations of the HTL – 'compact' and 'open' states – is likely due to varying interactions with lipids, including CL. In line with this hypothesis, augmenting the HTL with CL during purification increased the proportion of the 'open' state (from 8% to 17%), which could be enriched by glycerol gradient fractionation (to 32%), and further stabilised by cross-linking (to 40%) (*Figure 3—figure supplement 1*).

Evidently then, it seems likely that the open conformation (*Figure 3*, structure iv) is the state capable of interacting with the BAM complex (*Figure 3*, structure ii). The 'open' structure, and the 'compact' structure seen before (*Botte et al., 2016*), may reflect different functional states of the translocon. Presumably, the HTL would be closed when idle in the membrane and would be opened to various degrees depending on the associated cytosolic partners (e.g. ribosomes or SecA), periplasmic factors (chaperones, BAM, etc), and various substrates (e.g. globular, membrane, or β-barrels). Thus, it is not suprising that when free of the constraints of the membrane, and in the harsh environment of a detergent micelle, that these various states can be adopted – explaining the observed heterogeneity.

## Increasing the distance between inner- and outer-membranes weakens the HTL-BAM interaction

The dimensions of the HTL-BAM structure are sufficient to span roughly the distance between the inner- and outer-membranes, but only just. Thus, increasing the thickness of the periplasm might therefore be expected to stymie formation of HTL-BAM complexes, as previously observed for other trans-periplasmic complexes (*Asmar et al., 2017*; *Cohen et al., 2017*). To test this prediction, we increased the thickness of the periplasm by manipulating the width-determining lipoprotein Lpp, which separates the outer-membrane from the peptidoglycan layer. Increasing the length of *lpp* increases the width of the periplasm, from ~250 Å for wild type *lpp* to ~290 Å when an additional 21 residues are added to the resultant protein ($Lpp_{+21}$) (*Asmar et al., 2017*; *Figure 4a*).

The experiments described above (*Figure 1d,e*) were repeated: extracting total membranes in the presence of CL for IP by antibodies raised against SecG. Blotting for SecD and BamA then provided a measure for interactions within HTL and between HTL and BAM, respectively (*Figure 4b,c*; *Figure 4—figure supplement 1*). Consistent with our model, when the inter-membrane distance was increased, the integrity of the HTL in the inner-membrane was unaffected, but the recovery of HTL-BAM was reduced more than threefold (*Figure 4b,c*; *Figure 4—figure supplement 1*).

## PMF stimulation of protein translocation through the inner-membrane by SecA and SecYEG is not conferred by proton passage through SecD

It has been known for many years that SecDF plays a critical role in protein secretion. The results above show that the periplasmic domains of HTL, and in particular those of SecDF, mediate the recruitment of the BAM complex, likely to facilitate the onward journey of proteins to the outer-membrane. Therefore, we decided to re-evaluate the precise role and activity of this ancillary sub-complex. Experiments were established to investigate: (1) the role of SecDF in SecA dependent protein transport through the inner-membrane via SecYEG and (2) the consequences of its interaction with the BAM machinery for outer-membrane protein maturation. In particular, we set out to explore

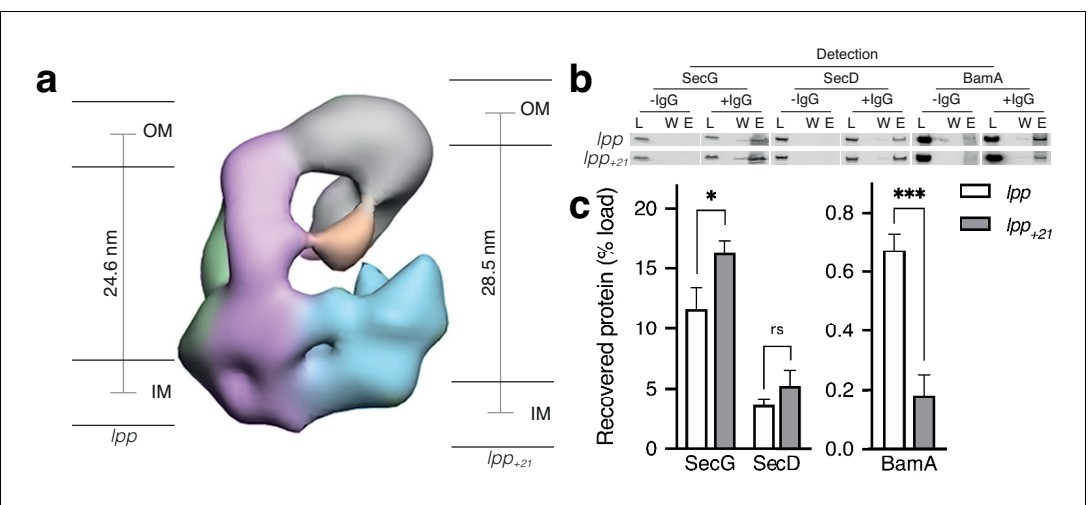

**Figure 4.** Effect of increasing periplasmic distance on the HTL-BAM interaction. (a) Negative-stain EM model of HTL-BAM (from *Figure 2a*), annotated with membranes at the experimentally determined distances between the inner- and outer- membranes of *E. coli* strains containing wild-type *lpp* and mutant *lpp+21* (*Asmar et al., 2017*). (b) Co-immunoprecipitation of SecG, SecD, and BamA when pulling from an anti-SecG monoclonal antibody. Co-IPs were conducted in the presence of cardiolipin as in *Figure 1d*, but with solubilised membranes of strains described in (a). (c) Quantification of IPs from (b). Error bars represent SEM. An unpaired T-test was used to compare samples (p=0.05, n = 3, * = <0.05, *** = <0.001, p values from left to right are 0.0170, 0.0990, and 0.0006).

The online version of this article includes the following figure supplement(s) for figure 4:

**Figure supplement 1.** Raw western blots of IPs investigating how periplasmic width effects the HTL-BAM interaction.

the possibility of an active role in these events for the proton translocating activity of the SecDF sub-complex.

*secDF* null mutants exhibit a severe export defect and are only just viable (*Pogliano and Beckwith, 1994*). To explore this phenotype further, we utilised *E. coli* strain JP325, wherein the expression of *secDF* is under the control of an *ara* promoter: the presence of arabinose or glucose results in production or depletion, respectively, of SecDF-YajC (*Economou et al., 1995*; *Figure 5a*; *Figure 5—figure supplement 1a*). To begin with, we grew cultures of JP325 containing either an empty vector, recombinant *secDF* or *secD$_{D519N}$F* overnight in permissive (0.2% arabinose) conditions. The following morning excess arabinose was washed away by centrifugation and resuspension, before

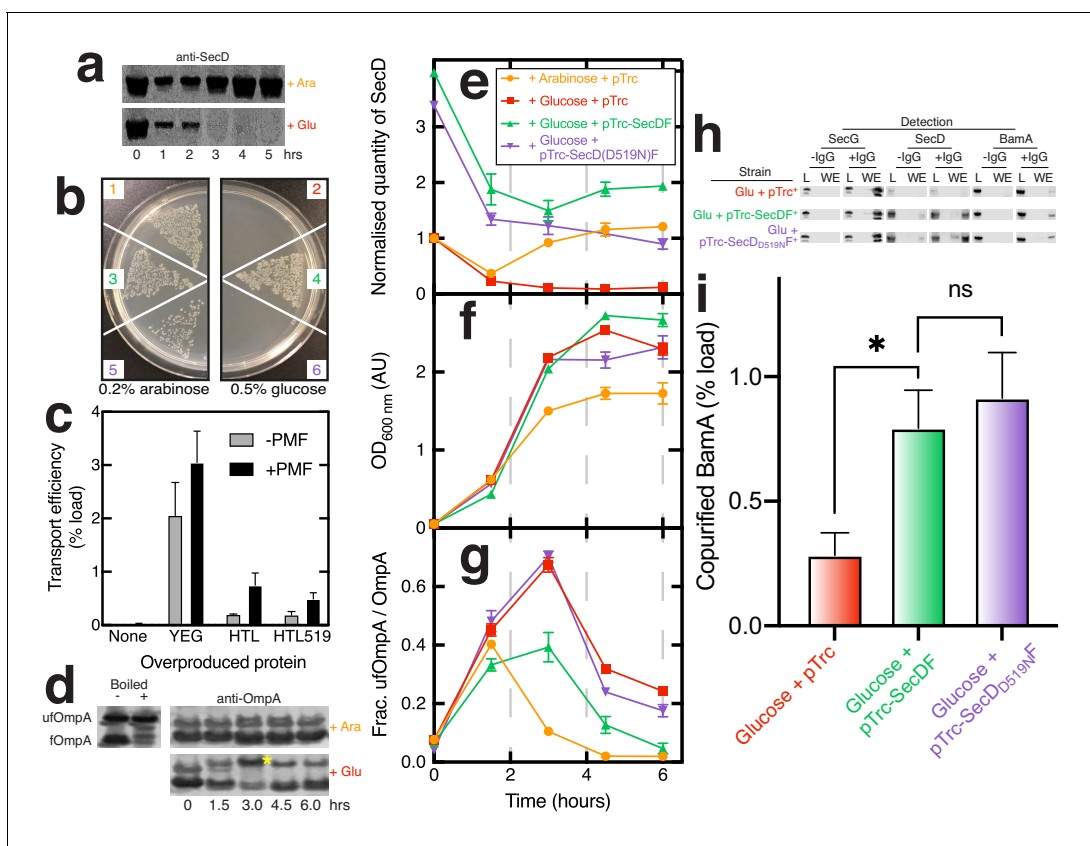

**Figure 5.** Effects of SecD depletion upon cell growth, OmpA transport across the inner-membrane, and maturation. (a) Western blot illustrating depletion of SecD in *E. coli* JP325 whole cells when grown in the presence of arabinose or glucose. t = 0 represents the time at which an overnight culture (grown in arabinose) was used to inoculate a secondary culture containing either arabinose or glucose. (b) Growth of *E. coli* JP325 transformed with empty vector (pTrc99a, 1 + 2), pTrc99a-*secDF* (3 + 4), and pTrc99A-*secD$_{D519N}$F* (5 + 6). Primary cultures were prepared in permissive conditions (arabinose). Cells were then washed and plated onto LB-arabinose (left panel) or LB-glucose (non-permissive, right panel). (c) Classical SecA-driven *in vitro* import assay with *E. coli* inverted inner-membrane vesicles (IMVs) and proOmpA. IMVs contained over-produced protein as stated on the x-axis. Error bars represent SEM (n = 3). (d) Periplasmic fractions of *E. coli* JP325 immunoblotted for OmpA. Folded OmpA (bottom band fOmpA) and unfolded OmpA (top band, yellow asterisk; ufOmpA) are shown. Also shown are control lanes containing *E. coli* whole cells with over-produced, mainly 'folded' OmpA (fOmpA, bottom band) and the same sample, but boiled, to produce 'unfolded' OmpA (ufOmpA, top band). For (e–g), samples were prepared from various cell cultures; see key (inset (e)) for strains used. Error bars represent SEM (n = 3 for experimental samples grown in glucose). (e) Quantification of SecD from western blots such as those shown in (a) (*Figure 5—figure supplement 1a*). Values are normalised to JP325-pTrc99a at t = 0. (f) Culture growth curves. (g) Analysis of western blots such as those from (d) and *Figure 5—figure supplement 1d* showing the quantity of ufOmpA as a fraction of the total OmpA in the periplasmic fraction. (h) Representative western blots of co-immuno-precipitations conducted as in *Figure 1d* in the presence of CL, but with solubilised membranes prepared from *E. coli* JP325 grown in the presence of glucose and cloned with variants of pTrc99a, as stated in the figure. (i) Quantification of BamA pull-down from co-IPs shown in (h). Error bars represent SEM. An unpaired T-test was used to compare samples (p=0.05, n = 3, * = <0.05, p values from left to right are 0.0449 and 0.6412).

The online version of this article includes the following figure supplement(s) for figure 5:

**Figure supplement 1.** Raw western blots and control quantifications accompanying SecD depletion experiments from *Figure 5*.

applying to plates containing either arabinose or glucose, for continued production or depletion of endogenous SecDF-YajC, respectively.

Depletion of SecDF-YajC results in a strong growth defect (*Figure 5b*, panels 1 and 2), which can be rescued by recombinant expression of *wild-type secDF* (*Nouwen and Driessen, 2002*; *Figure 5b*, panels 3 and 4). In contrast, expression of $secD_{D519N}F$, which results in the production of a complex incapable of proton transport (*Furukawa et al., 2017*), did not complement the defect (*Figure 5b*, panels 5 and 6). This phenotype is consistent with a general secretion defect, shown previously (*Gardel et al., 1987*).

In order to determine if this secretion defect is due to a problem in translocation through the inner-membrane (HTL), or beyond, we set up a classical *in vitro* transport assay: investigating SecA-driven proOmpA transport into inverted inner-membrane vesicles (IMVs) containing either over-produced native HTL, or the defective version of HTL (containing $SecD_{D519N}F$). Both sets of vesicles contained similar concentrations of SecY (*Figure 5—figure supplement 1b*), yet despite the blocked proton pathway through SecDF, there was little difference in the efficiencies of transport (*Figure 5c*). The lower quantities of transported pre-protein compared to experiments conducted with IMVs made from cells over-producing only the core-complex (SecYEG), seen also previously (*Schulze et al., 2014*), most likely reflects the reduced quantities of SecYEG in the IMVs made from HTL-producing cells, measured by blotting for SecY (*Figure 5—figure supplement 1b*).

Most importantly, the results demonstrate that SecA mediated ATP- and PMF-driven protein translocation through the inner-membrane via HTL does not require a functional proton wire through SecDF (*Figure 5c*). In this respect, SecYEG and SecA are sufficient (*Brundage et al., 1990*). Therefore, the proton translocating activity of SecD, needed for general secretion and cell survival, must be required for something downstream of protein transport through the inner-membrane.

## Interaction between the Sec and BAM complexes is required for efficient OmpA folding

The most obvious function of an interaction between the Sec and BAM machineries would be to facilitate efficient delivery and insertion of OMPs to the outer-membrane. We therefore reasoned that disrupting this interaction might compromise OMP delivery to BAM, leading to the accumulation of unfolded OMPs in the periplasm – particularly when high levels of outer-membrane biogenesis are required, such as in rapidly dividing cells.

Elevated levels of unfolded OmpA (ufOmpA) in the periplasm are a classical signature of OMP maturation deficiencies (*Sklar et al., 2007*; *Bulieris et al., 2003*). It can be easily monitored by SDS-PAGE and western blotting: folded OmpA (fOmpA) does not denature fully in SDS unless boiled; it therefore runs at a lower apparent molecular mass compared to ufOmpA when analysed by SDS-PAGE (*Figure 5d*, left; *Figure 5—figure supplement 1c*; *Sklar et al., 2007*; *Bulieris et al., 2003*). Importantly, we confirm the distinct identities of ufOmpA and fOmpA bands in the western blots by the analysis of native (folded) and boiled OmpA (unfolded). We also show the unfolded and folded forms also migrate differently from the precursor – proOmpA (*Figure 5—figure supplement 1c*). Therefore, the subsequent periplasmic analysis could not have been confused by un-secreted pre-protein – potentially from contaminating cytosol.

Based on the above results, SecDF looks like the most important mediator of the Sec-BAM interaction. We therefore used the SecDF depletion strain (JP325) as a basis for functional assays. To overcome the growth defect (*Figure 5b*, panels 1 and 2) and produce sufficient cells to analyse, overnight cultures of the strains used above were grown in permissive media (arabinose). Cells were then washed thoroughly to remove arabinose and transferred to new media containing glucose (non-permissive), or maintained in arabinose as a control, then resuspended to give an $OD_{600nm} = 0.05$ (marked as t = 0 in *Figure 5a,d,e–g*). Samples were taken from the growing cultures at regular intervals and the ratio of unfolded to folded OmpA determined (*Figure 5d,g*), along with cell density (*Figure 5f*) and SecD levels (*Figure 5e*). Under SecDF depletion conditions (*Figure 5e*, red squares), high levels of unfolded OmpA accumulate in the periplasm, particularly during the exponential phase when the demand for outer-membrane biogenesis is highest (*Figure 5d*, yellow asterisk; *Figure 5f,g*; *Figure 5—figure supplement 1d*). Meanwhile, under permissive conditions (*Figure 5e–g*, arabinose, orange circles), a more modest increase in ufOmpA is observable after 1.5 hr, but it recovers fully by 3 hr. Notably, this change is accompanied by a transient decrease in SecDF levels (*Figure 5e*, orange circles).

We know that these experiments were not compromised by the precursor proOmpA, which was not present in the periplasmic samples (*Figure 5—figure supplement 1c*). However, in some cases, a spurious band appeared in the OmpA western blots between the unfolded and folded forms (*Figure 5—figure supplement 1c*, red asterisk; *Figure 5—figure supplement 1d*). The band was only apparent in samples derived from overnight cultures grown in the presence of arabinose, including in the *wild-type* parent strain MC4100 (*Figure 5—figure supplement 1c*, far right lane, red asterisk). The stationary state of these cultures, grown in permissive and native conditions – with no impediment, or high demand for OmpA maturation – should not induce a build up of unfolded OmpA. So, it is unlikely that this spurious band represents an additional unfolded state of OmpA, and was ignored in the analysis.

Clearly, the expression of *secDF* and levels of ufOmpA in the cell envelope are anti-correlated, exacerbated during fast cell growth. These effects were not an indirect consequence of BamA loss, which was unperturbed (*Figure 5—figure supplement 1e*). Taken together, the data show that depletion of SecDF reduces the interaction between HTL and BAM, and thereby hampers transport of β-barrel proteins to the outer-membrane resulting in a build-up of ufOmpA in the periplasm. A backlog of unfolded OMP could compromise outer-membrane biogenesis and its integrity, and thereby explain the cold-sensitivity of *secDF* mutants (*Gardel et al., 1987*). This seems the most plausible explanation as transport through the inner membrane is unaffected by the absence of SecDF (*Schulze et al., 2014*; *Figure 5c*).

## Proton transport through SecD is required for efficient outer-membrane protein maturation

Proton translocation through SecD is crucial for cell growth (*Figure 5b*, panels 5 and 6), but evidently not for PMF-stimulated protein transport through the inner-membrane via SecYEG (*Figure 5c*). To determine if this activity is required for downstream events – such as delivery of OMP to the outer-membrane – we once again deployed the SecDF depletion strain, complemented with wild type or mutant *secDF* (as above, *Figure 5b*, panels 3–6), wherein the mutant produced SecD is incapable of proton transport (SecD$_{D519N}$).

Comparable quantities of the respective SecD variants could be produced (*Figure 5e*, green and purple; *Figure 5—figure supplement 1a*). The subsequent analysis showed the wild type, but not the mutant, reduced unfolded OmpA in the periplasm to levels much closer to that of the strain grown in permissive conditions (*Figure 5g*; green and purple, respectively; *Figure 5—figure supplement 1d*). Therefore, proton transport through SecD is apparently required for efficient outer-membrane protein folding.

To confirm the defective variant SecD$_{D519N}$F still interacts with BAM, we repeated co-IP experiments as before (*Figure 1d,e*) using membrane extracts derived from the SecDF depletion strains grown in the non-permissive condition (glucose; *Figure 5b*), but complemented with plasmids driving the expression of the wild type or mutant *secDF*, or nothing at all (empty plasmid). Again, in order to prepare sufficient material, overnight cultures were grown in media containing arabinose and then transferred to new media containing glucose. At OD$_{600nm}$ = 1.0, the cultures were harvested and membranes were prepared and solubilised for IP with SecG antibodies (*Figure 5h,i*; *Figure 5—figure supplement 1f,g*). As expected the immuno-precipitated yields of SecG were invariant, but the depletion of SecD (cells harbouring only the empty vector; *Figure 5—figure supplement 1g*) reduced the recovery of BamA commensurately (*Figure 5h,i*). The levels of co-immuno-precipitated SecG, SecD, and BamA were the same irrespective of complementation with the wild type or mutant forms of *secDF*. Evidently then, the integrity of the HTL and its ability to interact with the BAM complex do not require a functional proton wire through SecD. Therefore, the mutant's compromised OmpA maturation must be due to the loss of proton flow through SecD, rather than a loss of contact between HTL and BAM.

## HTL(SecD$_{D519N}$F) adopts a different conformation to the native version

The PMF-dependent mobility of the periplasmic domain of SecD (*Furukawa et al., 2017*) seems like it might be critical for its activity as part of the BAM-HTL complex. To test this, the variant of HTL containing SecD$_{D519N}$ was produced for comparison with the native form. Electron microscopy was used to assess the extent of 'compact' and 'open' forms of the HTL complex (*Figure 3*; *Figure 6*,

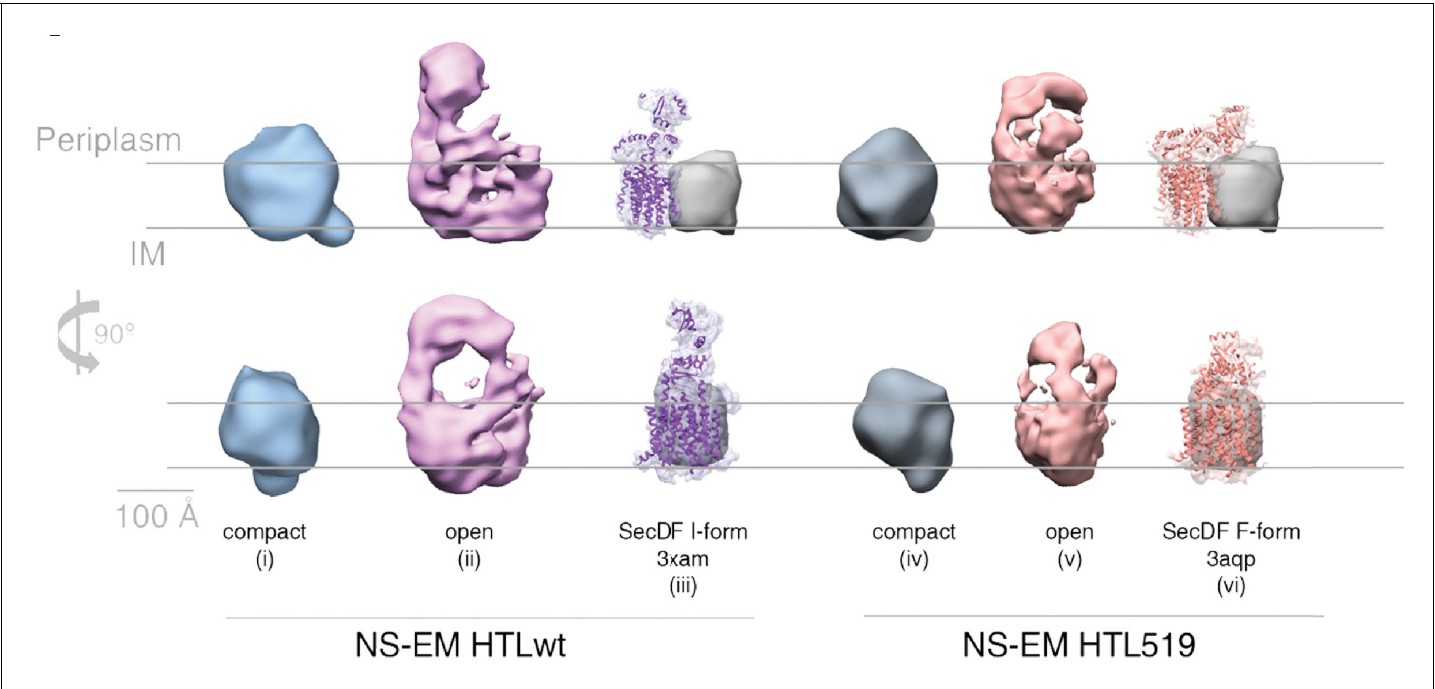

**Figure 6.** Structural comparison of HTL and HTL519. Comparison of negative stain-EM structures of 'compact' (structures i and iv) and 'open' (structures ii and v) conformations of HTL versus the counterpart containing SecD$_{D519N}$F (HTL519), both in the presence of CL. Atomic structures of SecDF overlaid with filtered maps at 5 Å are shown alongside for the *I*-form (structure iii, 3XAM) and *F*-form (structure vi, 3AQP), with the amino acid substitution equivalent to the *E. coli* SecD$_{D519N}$ in 3AQP. The grey arbitrary mass indicates the approximate position and mass of SecYEG. The online version of this article includes the following figure supplement(s) for figure 6:

**Figure supplement 1.** Negative-stain EM of the native HTL and the version containing SecD$_{D519N}$ (HTL519).

respectively, structures i and ii). The 2D classification of HTL-SecD$_{D519N}$ shows the open state is populated to a similar extent compared to the unmodified HTL (***Figure 6—figure supplement 1***).

The 3D analysis shows the compact states in both cases, similar to those seen before (***Botte et al., 2016***; ***Figure 3***, structure i; ***Figure 6***, structures i and iv). However, the 'open' states are significantly different: blocking the proton pathway in SecD results in a shorter extension of the periplasmic domains of the HTL, compared to the native version (***Figure 6***, structures ii versus v). This is consistent with the conformational change observed at atomic resolution in SecDF alone (***Figure 6***, structures iii versus vi) (***Furukawa et al., 2017***). Even at the current low-resolution description of the HTL-BAM complex (***Figure 2***), it is clear that these observed PMF-dependent conformational changes of SecD would be communicated to the outer-membrane.

## Discussion

The *in vivo* and *in vitro* analyses described here demonstrate a direct, functional interaction between the Sec and BAM translocons, mediated by the extended periplasmic domains possessed by BAM (***Ma et al., 2019***), SecDF (***Furukawa et al., 2017***) and YidC (***Kumazaki et al., 2015***), but not SecYEG (***Van den Berg et al., 2004***). Evidently, direct contact between HTL and BAM is required for efficient OMP biogenesis in rapidly growing cells. The interaction could enable large protein fluxes to stream through the periplasm, while minimising aggregation and proteolysis (***Figure 7***). The presence of super-complexes that bridge both membranes appears to be a fundamental feature of the Gram negative bacterial cell envelope – critical for a whole range of activities including the export of proteins through a gamut of different secretion systems (e.g. type I, II, III, IV, and VI) (***Green and Mecsas, 2016***); now including the Sec machinery. The general importance of these inter-membrane associations is only just coming to the fore (***Rassam et al., 2015***; ***Rassam et al., 2018***).

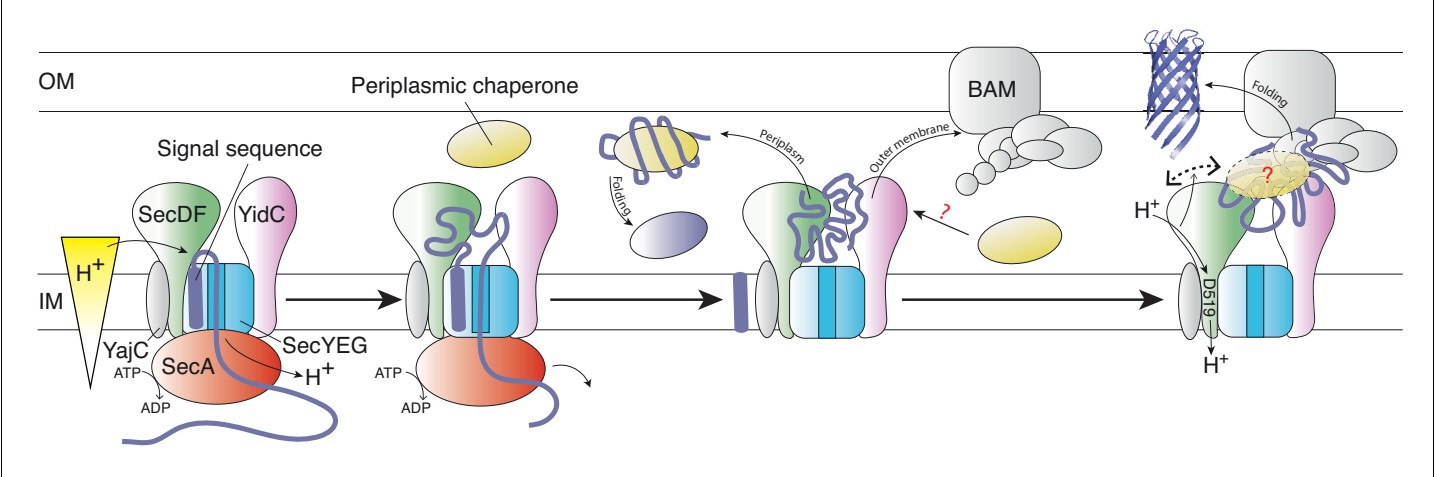

**Figure 7.** Schematic representation of the HTL-BAM machinery. Model of OMP transfer through the bacterial envelope, facilitated by HTL-BAM and periplasmic chaperones, such as SurA, Skp, PpiD, and YfgM. From left to right: OMP precursors with an N-terminal signal sequence are driven across the membrane by the ATPase SecA through the Sec translocon – this process is stimulated by the PMF (independent of SecDF). Late in this process, the pre-protein emerges into the periplasm and the signal sequence is removed, releasing the mature protein. Presumably, globular proteins are then guided into the periplasm, where folding will occur assisted by periplasmic chaperones. Otherwise, OMP-chaperone-HTL complexes are recognised by the BAM complex, with interactions forming between BAM and both HTL (this study) and SurA (*Sklar et al., 2007*). The persistence and variety of chaperones involvement at this stage is unclear (?). This conjunction enables the smooth and efficient passage of OMPs to the outer-membrane, which is enabled by coupling of the inner-membrane proton-motive force with conformational changes in the periplasmic domain of SecDF (right).

It has already been shown that the HTL encloses a lipid-containing cavity within the membrane, presumably to facilitate membrane protein insertion (*Botte et al., 2016*; *Martin et al., 2019*). Remarkably, in the super-complex between HTL and BAM there is a much larger extension of this cavity opening into the periplasm (*Figure 2*). This would seem an obvious place for OMP passage and for the interaction with chaperones (*Figure 7*) and is of sufficient size to do so. The cavity is situated such that a secretory protein could enter via the protein-channel through SecYEG, and then exit accordingly into the periplasm, or into the mouth of the BAM complex.

It remains to be seen how the Sec-BAM complex and the periplasmic chaperones coordinate. Perhaps these chaperones recognise emerging globular proteins at the Sec-machinery and shuttle them into the periplasm, with or without the need for the BAM complex. Otherwise, they could facilitate passage of OMPs through the inter-membrane assembly for outer-membrane folding and insertion by BAM (*Figure 7*). SurA is known to interact with BamA (*Sklar et al., 2007*), and an interaction with the HTL also seems likely (*Figure 7*). Other ancillary factors of the Sec machinery have also been implicated: YfgM and PpiD are thought to mediate interactions between emergent periplasmic proteins and chaperones (*Götzke et al., 2014*); indeed, PpiD has also been shown to interact with SecYEG and YidC (*Jauss et al., 2019*). Interestingly, *yfgL* and *yfgM* are in the same operon (*Blattner et al., 1997*), the former encoding a subunit of the BAM complex (BamB) (*Wu et al., 2005*). Moreover, a recent proteomic analysis of the *E. coli* 'membrane protein interactome' identifies cross-membrane interactions involving SecYEG, BAM and the chaperones YfgM and PpiD (*Carlson et al., 2019*). Clearly, understanding the interplay of various periplasmic chaperones during OMP passage through the Sec-BAM assembly to the outer-membrane will require further attention.

From our data it is clear that the periplasmic domain of SecD is central to the physical HTL-BAM interaction. Even more intriguing though is the requirement for a functioning proton wire through the SecDF trans-membrane domain. The non-functioning SecDF is fully capable of conferring an interaction with BAM but is presumably unable to transmit PMF-dependent conformational changes relayed from the inner-membrane. This static interaction of HTL and BAM is insufficient to enable efficient OMP maturation. The consequences of preventing PMF inducing dynamic interplay between HTL and BAM are as severe as the disconnection induced by SecDF depletion. Presumably, the

deletion of the periplasmic domain P1 of SecD, which also eliminates the interaction with the outer-membrane, will have an equally severe effect.

The requirement for PMF-driven inter-membrane dynamic connectivity raises the intriguing prospect of TonB-style energy-coupling from the inner-membrane (*Celia et al., 2016*): that is the transmission of free energy available from the PMF via the Sec-machinery (*Brundage et al., 1990*; *Arkowitz and Wickner, 1994*; *Schiebel et al., 1991*) for OMP folding and insertion at the outer-membrane. We therefore propose that one of the primary roles of SecDF is in inter-membrane trafficking and energy transduction. Indeed, we and others have shown that ATP- and PMF-driven transport of proteins through the inner-membrane is dependent only on SecYEG and SecA (*Brundage et al., 1990*; *Schulze et al., 2014*), whereas we show here that proton translocation through SecD is crucial for efficient OMP folding and growth.

Thus, there appears to be two distinct requirements of the PMF in protein secretion: one for the early stage – SecA-driven translocation through SecYEG at the inner-membrane, and another for late stages of OMP maturation. The latter is facilitated by conformational changes in SecDF for transduction of energy from the inner- to the outer-membrane. Here, we show that an 'open' state of the HTL interacts with the BAM complex, and that the periplasmic regions of SecD can adopt different conformations, reminiscent of those previously characterised as the *I*- and *F*-forms (*Furukawa et al., 2017*); when a key proton carrying residue of the inner-membrane segment of the translocon is neutralised – $SecD_{D519N}$ – the periplasmic domain adopts the *F*-form (*Furukawa et al., 2017*). Thus, successive protonation (approximated by $SecD_{D519N}$) and deprotonation result in large, cyclical movements – between the *I*- and *F*-forms – during PMF-driven proton transport from the periplasm to the cytosol (*Figure 7*). Presumably then, the occurance of these conformational changes, while connected to the BAM complex, results in long-range energetic coupling between the inner- and outer-membranes. Interestingly, the phospholipid cardiolipin (CL) is important for the stabilisation of the 'open' state of the HTL and its interaction with the BAM machinery. It is probably not a coincidence that this lipid has already been shown to be critical for PMF-driven protein translocation through SecYEG (*Corey et al., 2018*). Certainly, we hope to overcome the inherent flexibility of the CL-stabilised open translocon, primed to receive BAM, in order to determine its high-resolution structure, and further understand this process.

Taking all together, this builds a compelling case for SecD mediated inter-membrane energy transduction – in keeping with other members of the RND transporter family, such as the assembly of AcrAB (inner-membrane) and TolC (outer-membrane) (*Du et al., 2014*; *Wang et al., 2017*). Direct association between inner- and outer-membrane components appears to be the rule rather than the exception for transporters embedded in double membrane systems: parallels with the translocation assembly module (TAM) for auto-transporter secretion (*Selkrig et al., 2012*), and the TIC-TOC import machinery of chloroplasts (*Chen et al., 2018*) are striking, given the respective outer-membrane components (TamA and TOC75) are homologous of BamA. Particularly intriguing is the possibility of the mitochondrial homologue of BAM (sorting and assembly machinery; SAM) participating in analogous inter-membrane interactions between inner- and outer-membranes. Indeed, subunits of the MItochondrial contact site and Cristae Organizing System (MICOS) connect the energy-transducing ATP synthase of the inner-membrane and SAM at the outer-membrane (*Ott et al., 2015*; *Rampelt et al., 2017*).

## Materials and methods

### Strains and plasmids

*E. coli* C43 (DE3) was a gift from Sir John Walker (MRC Mitochondrial Biology Unit, Cambridge, UK) (*Miroux and Walker, 1996*). *E. coli* BL21 (DE3) were purchased as competent cells (New England Biolabs). *E. coli* Δ*secG* (KN425 (W3110 M25 Δ*secG*::kan)) (*Nishiyama et al., 1994*), which lacks a genomic copy of *secG*, was obtained from Prof. Frank Duong (University of British Colombia, Vancouver, Canada). *E. coli* strain jp325 (Kan^r), which contains an arabinose-regulated *secDF-yajC* operon (*Economou et al., 1995*), was given to us by Prof. Ross Dalbey.

The plasmids for over-expression of *secEYG* and *yidC* were from our laboratory collection (*Collinson et al., 2001*; *Lotz et al., 2008*), the former and also that of *secDF* were acquired from Prof. Frank Duong (*Duong and Wickner, 1997*). Vectors designed for over-production of HTL, HTL

($\Delta$YidC) and HTL(SecD$_{D519N}$) were created using the ACEMBL expression systems (*Botte et al., 2016*; *Bieniossek et al., 2009*). The vector for *bamABCDE* over-expression pJH114 (Amp$^r$) was a gift from Prof. Harris Bernstein (*Roman-Hernandez et al., 2014*) from which pJH114-*bamACDE* ($\Delta$BamB) was produced by linear PCR with primers designed to flank the BamB gene and amplify DNA around it. FseI restriction sites were included in the primers to ligate the amplified DNA. pBAD-SecDF($\Delta$P1) was generated by amplifying SecDF($\Delta$P1) from pBAD-SecDF and cloning it between the pBAD NcoI and HindIII sites (*Komar et al., 2016*).

For SecDF depletion experiments, SecDF was cloned into pTrc99a (Amp$^r$, IPTG-inducible), and the *secD$_{D519N}$* mutation was subsequently made by changing the wild type carrying plasmid using a site-directed ligase-independent PCR method.

## SDS-PAGE, western blotting, and antibodies

All SDS-PAGE was performed with either *Invitrogen* Bolt 4–12% Bis-Tris gels or *Invitrogen* midi 4–12% Bis-Tris gels. For western blotting, proteins were transferred onto nitrocellulose membrane. Mouse monoclonal antibodies against SecY, SecE and SecG were from our laboratory collection (used at 1:10000 dilution). Polyclonal antibodies against SecD and BamA were generated commercially in rabbits (all used at 1:5000 dilution). BamB and BamD antibodies were gifts from Dr Harris Bertstein (1:5000 dilution). A secondary antibody conjugated to DyLight800 was used for SecG and SecY (Thermo Fisher Scientific, 1:10000 dilution), whereas a HRP-conjugated secondary antibody was used for SecD and BamA (1:10000 dilution).

## Protein production and purification

HTL, HTL($\Delta$YidC), HTL(SecD$_{D519N}$), SecYEG, YidC, and SecDF were purified as described previously (*Schulze et al., 2014*; *Collinson et al., 2001*; *Lotz et al., 2008*; *Burmann et al., 2013*). BAM and BAM($\Delta$BamB) was over-produced in *E. coli* C43 according to established protocols (*Iadanza et al., 2016*; *Roman-Hernandez et al., 2014*; *Kessner et al., 2008*).

## Isolation of inner and outer membranes

One litre of *E. coli* cultures over-producing SecYEG, HTL, SecDF, or SecDF($\Delta$P1) were produced as described previously (*Schulze et al., 2014*; *Collinson et al., 2001*; *Komar et al., 2016*). The harvested cell pellets were resuspended in 20 mL TS$_{130}$G, homogenised with a potter, passed twice through a cell disruptor (Constant Systems Ltd.) for lysis and centrifuged to remove debris (SS34 rotor, Sorvall, 12000 x*g*, 20 min, 4˚C). The supernatant was taken and layered upon 20 mL TS$_{130}$G + 20% (w/v) sucrose in a Ti45 tube and centrifuged (Ti45 rotor, Beckmann-Coulter, 167000 x*g*, 120 min, 4˚C). The pellet was taken, resuspended in 4 mL TS$_{130}$G, homogenised with a potter and layered upon a sucrose gradient prepared in an SW32 centrifuge tube composed of 5 mL layers of TS$_{130}$G + 55% (w/v), 48%, 41%, 34%, and 28% sucrose. The sample was then fractionated by centrifugation (SW32 rotor, Beckmann-Coulter, 130000 x*g*, 15 hr, 4˚C). Upon completion, the light to heavy fractions were analysed by SDS-PAGE and western blotting.

## Co-immunoprecipitations (co-IPs) with *E. coli* total membrane extracts

Membrane pellets of *E. coli* strains C43 (WT), C43 pJH114-*bamABCDE* (Amp$^r$), $\Delta$*secG* (Kan$^r$), WT *lpp*, mutant *lpp$_{+21}$* and JP325 (containing variants of pTrc as specified in text, cultures grown in glucose for depletion of endogenous SecDFyajC), were prepared as described previously (*Collinson et al., 2001*), with *bamABCDE* over-expression achieved as before (*Roman-Hernandez et al., 2014*). The pellets were resuspended in TS$_{130}$G to 120 mg/mL, homogenised and solubilised with 0.5% DDM for 1 hr at 4˚C. The solubilised material was clarified by ultra-centrifugation (160000 x*g* for 45 min) and the membrane extracts were analysed.

For co-IPs pulling on SecG antibody, 250 µL of protein G resin was washed in a spin column with 200 mM NaCl, 20 mM HEPES pH 8.0 (HS buffer), and blocked overnight in HS buffer + 2% (w/v) BSA at 4˚C. Meanwhile, 7.5 µL of purified SecG monoclonal antibody was added to 500 µL of the membrane extracts and incubated overnight at 4˚C. The following morning, the resin was washed thoroughly in HS buffer containing either 0.02% (w/v) DDM or 0.02% (w/v) DDM with 0.002% (w/v) CL, resuspended back to 250 µL and added to the 500 µL of membrane extract and IgG mixture for three hours rotating gently at room temperature. The resin was separated from the extracts by

centrifugation in a spin column at 500 x*g* for 1 min, washed six times with 350 µL HS buffer, followed by one final wash with 150 µL HS buffer, which was collected in a fresh tube for analysis (to which 50 µL of 4x LDS sample buffer was added once collected). The bound material was then eluted by addition of 150 µL 1 x LDS sample buffer (to which an additional 50 µL of 1x LDS sample buffer was added once collected). Samples were analysed by SDS-PAGE and western blotting.

For co-affinity adsorption by pulling on the hexa-histidine tag of recombinant BamA, 100 µL of nickel-charged chelating resin was added to 500 µL of membrane extracts and incubated for 5 min at room temperature. The resin was then separated from the extract and treated in the same way as described above but with $TS_{130}G$ + 0.02% (w/v) DDM + 0.002% (w/v) CL + 30 mM imidazole (washing) or 300 mM imidazole (elution).

Statistical analyses were conducted using GraphPad Prism. An unpaired T-test was used to compare pull-down samples (p-value=0.05, * = <0.05, ** = <0.01, *** = <0.001, specific p-values are stated in figure legends).

## *In vitro* assembly and purification of complexes for EM and XL-MS

All protein complexes visualised by negative stain EM were formed by incubating 5 µM of the respective proteins in binding buffer (20 mM HEPES pH 8.0, 250 mM NaCl, 0.03% (w/v) DDM, 0.003% (w/v) CL at 30°C for 30 min with shaking in a total volume of 150 µL. The protein complexes were purified in a glycerol/glutaraldehyde gradient (20–40% (w/v) and 0–0.15% (w/v), respectively) by centrifugation at 34000 RPM in a SW60 Ti rotor (Beckmann-Coulter) for 16 hr at 4°C. Mobility controls of individual and partial complexes (BAM, and HTL) or individual (SecYEG, YidC, SecDF) without the glutaraldehyde gradient were performed under the same conditions. Gradients were fractionated in 150 µL aliquots and those with glutaraldehyde were inactivated with 50 mM of Tris pH 8.0. Aliquots were analysed by SDS-PAGE and silver staining.

The HTL-BAM complex for cryo-EM was formed by incubating 8 µM of the HTL and BAM complexes in binding buffer (50 mM HEPES pH 8.0, 200 mM NaCl, 0.01% (w/v) DDM/0.001% (w/v) CL) at 30°C for 20 min with shaking in a total volume of 250 µL. After 20 min, 0.05% of glutaraldehyde was added to the sample and incubated for 10 min at 21°C. The crosslinker was inactivated with 30 mM Tris pH 8.0 and the sample was loaded onto a Superose 6 Increase 10/300 GL (GE healthcare) column equilibrated in GF buffer (30 mM Tris pH 8.0, 200 mM NaCl, 0.01% (w/v) DDM). Fractions were analysed by SDS-PAGE and silver staining.

The HTL-BAM complex for cross-linked mass spectroscopy (XL-MS) analysis was prepared following the same procedure described for the cryo-EM preparation, but the sample was crosslinked with 1.5 mM DSBU and inactivated with 20 mM of ammonium carbonate pH 8.0 before being loaded onto the gel filtration column.

## XL-MS analysis

The DSBU cross-linked HTL-BAM complex was precipitated by methanol and chloroform (*Wessel and Flügge, 1984*) and the pellet dissolved in 8 M urea. After reduction with 10 mM DTT (1 hr at 37°C) and alkylation with 50 mM iodoacetamide (30 min in the dark at RT), the sample was diluted 1:5 with 62.5 mM ammonium hydrogen carbonate and digested with trypsin (1:20 w/w) overnight at 37°C. Digestion was stopped by the addition of formic acid to a final concentration of 2% (v/v) and the sample split in two equal amounts for fractionation by size exclusion (SEC) and reverse phase C18 at high pH chromatography. A Superdex Peptide 3.2/300 column (GE Healthcare) was used for SEC fractionation by isocratic elution with 30% (v/v) acetonitrile/0.1% (v/v) TFA at a flow rate of 50 µL/min. Fractions were collected every minute from 1.0 mL to 1.7 mL of elution volume. Reverse phase C18 high pH fractionation was carried out on an Acquity UPLC CSH C18 1.7 µm, 1.0 × 100 mm column (Waters) over a gradient of acetonitrile 2–40% (v/v) and ammonium hydrogen bicarbonate 100 mM.

All the fractions were lyophilised and resuspended in 2% (v/v) acetonitrile and 2% (v/v) formic acid for LC–MS/MS analysis. An Ultimate U3000 HPLC (ThermoScientific Dionex, USA) was used to deliver a flow of approximately 300 nL/min. A C18 Acclaim PepMap100 5 µm, 100 µm × 20 mm nanoViper (ThermoScientific Dionex, USA), trapped the peptides before separation on a C18 Acclaim PepMap100 3 µm, 75 µm × 250 mm nanoViper (ThermoScientific Dionex, USA). Peptides were eluted with a gradient of acetonitrile. The analytical column was directly interfaced via a nano-

flow electrospray ionisation source, with a hybrid quadrupole orbitrap mass spectrometer (Q-Exactive HF-X, ThermoScientific, USA). MS data were acquired in data-dependent mode. High-resolution full scans (R = 120000, m/z 350–2000) were recorded in the Orbitrap and after CID activation (stepped collision energy 30 ± 3) of the 10 most intense MS peaks, MS/MS scans (R = 45,000) were acquired.

For data analysis, Xcalibur raw files were converted into the MGF format through MSConvert (Proteowizard; *Kessner et al., 2008*) and used directly as input files for MeroX (*Götze et al., 2015*). Searches were performed against an ad hoc protein database containing the sequences of the complexes and a set of randomised decoy sequences generated by the software. The following parameters were set for the searches: maximum number of missed cleavages 3; targeted residues K; minimum peptide length five amino acids; variable modifications: carbamidomethyl-Cys (mass shift 57.02146 Da), Met-oxidation (mass shift 15.99491 Da); DSBU modification fragments: 85.05276 Da and 111.03203 (precision: five ppm MS [*Kessner et al., 2008*] and 10 ppm MS [*Götze et al., 2015*]); False Discovery Rate cut-off: 5%. Finally, each fragmentation spectra were manually inspected and validated.

## EM and image processing

For negative stain EM, aliquots of sucrose gradient fractions containing the different complexes were applied to glow-discharged (15 s) carbon grids with Cu 300 mesh, washed and stained with 2% (w/v) uranyl acetate (1 min). Digital images were acquired with two different microscopes; a Tecnai 12 with a Ceta 16M camera (ThermoFisher Scientific) at a digital magnification of 49000 x and a sampling resolution of 2.04 Å per pixel, and in a Tecnai 12 with a Gatan Camera One View at a digital magnification of 59400 x and a sampling resolution of 2.1 Å per pixel. Image processing was performed using the EM software framework Scipion v1.2 (*de la Rosa-Trevín et al., 2016*). Several thousand particles were manually and semi-automatically supervised selected as input for automatic particle picking through the XMIPP3 package (*Abrishami et al., 2013*; *de la Rosa-Trevín et al., 2013*). Particles were then extracted with the Relion v2.1 package (*Scheres, 2012a*; *Kimanius et al., 2016*) and classified with a free-pattern maximum-likelihood method (Relion 2D-classification). After manually removing low-quality 2D classes, a second round of 2D classification was performed with Relion and XMIPP-CL2D in parallel (*Sorzano et al., 2010*). Representative 2D averages were used to generate several initial 3D models with the EMAN v2.12 software (*Scheres, 2012b*; *Tang et al., 2007*). Extensive rounds of 3D classification were then carried out using Relion 3D-classification due to the heterogeneity of the sample. The most consistent models were used for subsequent 3D classifications. For the final 3D volume refinement, Relion auto-refine or XMIPP3-Projection Matching were used. Resolution was estimated with Fourier shell correlation using 0.143 correlation coefficient criteria (*Rosenthal and Henderson, 2003*; *Scheres and Chen, 2012*). See *Figure 2—source data 1* for image processing details.

For cryo-EM, appropriate fractions of the glutaraldehyde-crosslinked HTL-BAM complex purified by gel filtration were applied to glow-discharged (20 s) Quantifoil grids (R1.2/R1.3, Cu 300 mesh) with an ultrathin carbon layer (2 nm), blotted and plunged into a liquid ethane chamber in a Leica EM GP. Two data sets from the same grid were acquired in a FEI Talos Arctica cryo-electron microscope operated at 200 kV and equipped with a K2 detector at calibrated magnification of 79000 x. The first data set with 2056 images recorded, had a 1.75 Å/px sample resolution, dose rate of 2.26 electrons/Å$^2$ and 20 s exposure time fractionated in 40 frames. Defocus values oscillated between −1.5 nm and −3.0 nm. The second data set with 3703 images recorded, had a 0.875 Å/px sample resolution, dose rate of 2.47 electrons/Å$^2$ and 18 s exposure time fractionated in 40 frames. Defocus values oscillated between – 1.0 nm and −2.2 nm. Particles were picked in the same way as for negative stain, and were binned to a 1.75 Å/px sample resolution before merging to the first data set. Image processing was performed using the EM software framework Scipion v1.2 (*de la Rosa-Trevín et al., 2016*) with a similar strategy to the negative stain-EM samples but also using extensive masking procedures (*Figure 2—figure supplement 4*).

All 3D reconstructions were calculated using a home-built workstation (CPU Intel Core i7 7820X, 2x Asus Turbo GTX 1080Ti, 16 Gb RAM DDR4) and partial usage of HPC clusters (Bluecrystal four and Bluecryo) at the University of Bristol.

## Depletion of SecDF-YajC

*E. coli* strain jp325 was transformed with empty pTrc99a, or the same plasmid, but cloned with either wild type *secDF* or s*ecD$_{D519N}$F*. Precultures of the strains were prepared in 100 mL 2xYT media supplemented with 0.2% (w/v) arabinose, ampicillin (100 µg/mL, for pTrc selectivity) and kanamycin (50 µg/mL, for jp325 selectivity). The following morning, the cells were harvested by centrifugation and resuspended with 50 mL fresh 2xYT (no arabinose). This washing procedure was repeated two more times to remove excess arabinose. Prewarmed (37˚C) 1 L 2xYT cultures containing either 0.2% (w/v) arabinose or 0.5% (w/v) glucose were then inoculated with the preculture such that a final $OD_{600\ nm}$ of 0.05 was achieved. An aliquot was taken every 1.5 hr for 6 hr. Induction of pTrc with IPTG was not necessary as background expression was sufficient to achieve levels of SecD similar to that of JP325 cultured in the presence of arabinose. Periplasmic fractions were produced by preparing spheroplasts (*Birdsell and Cota-Robles, 1967*), centrifuging the samples at 12000 x*g* for 5 min, taking the supernatant (a mixture of periplasmic and OM fractions) and removing the OM fraction by ultracentrifugation at 160000 x*g* for 20 min. The fractions were then subjected to SDS-PAGE and western blotting.

## Measurement of protein transport

Inner-membrane vesicles (IMVs) were produced from BL21(DE3) cells overproducing HTL, HTL (SecD$_{D519N}$), SecYEG or with empty pBAD as described previously (*Corey et al., 2018*). Transport experiments with and without PMF were performed in triplicate using established methods (*Corey et al., 2018*).

## Acknowledgements

We are particularly grateful for the generosity of Dr Harris Bernstein for the kind gifts of the *bamABCDE* expression construct (pJH114) and antibodies. Thanks to Prof. Daniel Daley for telling us about YfgL and YfgM. We thank Dr Remy Martin for brewing, and for making the HTL *secD$_{D519N}$F* mutant. We acknowledge access and support of the Wolfson Bioimaging Facility and the GW4 Facility for High-Resolution Electron Cryo-Microscopy, with particular thanks to Dr Ufuk Borucu. We are grateful to Mr Tom Batstone for the support at the computer cluster BlueCryo. The GW4 Facility for High-Resolution Electron Cryo-Microscopy was funded by the Wellcome Trust (electron microscope with direct electron detector; 202904/Z/16/Z and 206181/Z/17/Z) and BBSRC (computer cluster; BB/R000484/1).

## Additional information

### Funding

| Funder | Grant reference number | Author |
|---|---|---|
| Biotechnology and Biological Sciences Research Council | BB/S008349/1 | Sara Alvira<br>Daniel W Watkins<br>Ian Collinson |
| Biotechnology and Biological Sciences Research Council | BB/N015126/1 | Daniel W Watkins<br>Ian Collinson |
| Biotechnology and Biological Sciences Research Council | BB/M003604/1 | Sara Alvira<br>Ian Collinson |
| EMBO | ALTF 710-2015 | Sara Alvira |
| EMBO | LTFCOFUND2013 | Sara Alvira |
| EMBO | GA-2013-609409 | Sara Alvira |
| Elizabeth Blackwell Institute for Health Research | | Sara Alvira |
| Biotechnology and Biological Sciences Research Council | BB/M009122/1 | Luca A Troman |
| Biotechnology and Biological Sciences Research Council | BB/J014400/1 | Luca A Troman |

| Wellcome | 202904/Z/16/Z | Ian Collinson |
|---|---|---|
| Biotechnology and Biological Sciences Research Council | BB/R000484/1 | Ian Collinson |
| Wellcome | 206181/Z/17/Z | Bertram Daum<br>Vicki AM Gold |

The funders had no role in study design, data collection and interpretation, or the decision to submit the work for publication.

### Author contributions

Sara Alvira, Daniel W Watkins, Ian Collinson, Conceptualization, Resources, Data curation, Software, Formal analysis, Supervision, Funding acquisition, Validation, Investigation, Visualization, Methodology, Writing - original draft, Project administration, Writing - review and editing; Luca A Troman, Conceptualization, Data curation, Formal analysis, Validation, Investigation, Visualization, Methodology, Writing - review and editing; William J Allen, Conceptualization, Formal analysis, Investigation, Methodology, Writing - review and editing; James S Lorriman, Data curation, Formal analysis, Investigation, Methodology; Gianluca Degliesposti, J Mark Skehel, Resources, Data curation, Formal analysis, Investigation, Visualization, Methodology; Eli J Cohen, Morgan Beeby, Resources, Investigation, Methodology; Bertram Daum, Vicki AM Gold, Visualization, Writing - review and editing, Funding acquisition

### Author ORCIDs

Daniel W Watkins (iD) https://orcid.org/0000-0003-3825-5036
William J Allen (iD) http://orcid.org/0000-0002-9513-4786
James S Lorriman (iD) https://orcid.org/0000-0002-1755-0805
Morgan Beeby (iD) http://orcid.org/0000-0001-6413-9835
Bertram Daum (iD) https://orcid.org/0000-0002-3767-264X
Vicki AM Gold (iD) https://orcid.org/0000-0002-6908-0745
Ian Collinson (iD) https://orcid.org/0000-0002-3931-0503

### Decision letter and Author response

Decision letter https://doi.org/10.7554/eLife.60669.sa1
Author response https://doi.org/10.7554/eLife.60669.sa2

## Additional files

### Supplementary files

• Transparent reporting form

### Data availability

All data generated or analysed during this study are included in the manuscript and supplementary information. Information regarding statistical testing is located in materials and methods and corresponding figure legends.

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
