## [Decision Letter]

**Acceptance summary:**

The manuscript by Alvira et al. describes the structural investigation of the potential interaction between the Sec and BAM translocons that span the periplasmic space of gram negative bacteria. The authors show by immunoprecipitation, cross-linking and electron microscopy that both complexes interact with each other. In particular the cross linking / mass spectrometry data allow them to map the interaction between different components that confirm the model based on the EM data.

These investigations enhance our understanding of transport processes that include two membranes in bacteria.

**Decision letter after peer review:**

Thank you for submitting your article "Inter-membrane association of the Sec and BAM translocons for bacterial outer-membrane biogenesis" for consideration by *eLife*. Your article has been reviewed by two peer reviewers, and the evaluation has been overseen by Volker Dötsch as the Reviewing Editor and John Kuriyan as the Senior Editor.

The reviewers have discussed the reviews with one another and the Reviewing Editor has drafted this decision to help you prepare a revised submission.

Summary:

The manuscript by Alvira et al. descibes the structural investigation of the potential interaction between the Sec and BAM translocons that span the periplasmic space of gram negative bacteria. The authors show by immunoprecipitation, cross-linking and electron microscopy that both complexes interact with each other. In particular the cross linking / mass spectrometry data allow them to map the interaction between different components that confirm the model based on the EM data.

They further show that the HTL subcomplex adopts different conformations (open and closed) and that proton translocation via the SecDF subcomponent is important for folding of OmpA.

1) The western blot in Figure 5D contains several OmpA bands, but it does not indicate which of these bands is precursor OmpA. In addition, there is no control to establish which band is precursor OmpA. In Figure 5—figure supplement 1D, lanes 1-5, I see three bands: a high MW band (band 1), a middle MW band (band 2) and a lower MW band (band 3). To me, it appears that band 1 chases to band 2 and then to band 3, so I would guess that band 1 is precursor, band 2 is ufOmpA and band 3 is fOmpA. However, the arrows to the side of the blot suggest that the authors believe band 1 is ufOmpA. How can they be sure? How do the authors know that the indicated band in Figure 5D and Figure 5—figure supplement 1D is mature periplasmic ufOmpA and not precursor OmpA? Without this information it is not possible to make any conclusions about the secretion or assembly of OmpA.

2) The mutant SecD protein that lacks the periplasmic domain does not interact with the BAM machinery. Does this mutant behave similarly to the SecD(D519N) mutant in translocation assays?

3) The SurA data suggests a function for SecDF in bacteria that lack the BAM machinery, such as Gram-positive bacteria. However, these results are still very preliminary. The author should provide additional experiments demonstrating that the interaction between SecDF and SurA-OmpA is not an interaction artefact. Alternatively, because this interaction is not required for any of the authors' main conclusions, it could be removed from the paper.

4) More detailed structural characterization would of course be useful in describing the molecular mechanism but the current state convincingly shows the interaction between both translocon complexes. Have the authors tried to further characterize the open and closed states of the HTL complex by cross linking / mass spectrometry (for example before CL addition and glycerol gradient fractionation and after?

Revisions expected in follow-up work:

1) SecD was originally identified because mutants have a translocation defect. For example, in one of the papers cited by the authors, Beckwith and colleagues identified the secD locus based on the ability of mutations at this locus to cause defects in translocation. It would therefore be a little surprising if the SecD(D519N) mutant does not cause a translocation defect. If the authors wish to conclude that the SecD(D519N) mutant does not cause a defect in translocation, they should use a more sensitive assay (such as a pulse-chase assay). The assay used by authors measures steady state levels of protein, which could potentially miss transient translocation defects.

---

## [Author Response]

Revisions for this paper:1) The western blot in Figure 5D contains several OmpA bands, but it does not indicate which of these bands is precursor OmpA. In addition, there is no control to establish which band is precursor OmpA. In Figure 5—figure supplement 1D, lanes 1-5, I see three bands: a high MW band (band 1), a middle MW band (band 2) and a lower MW band (band 3). To me, it appears that band 1 chases to band 2 and then to band 3, so I would guess that band 1 is precursor, band 2 is ufOmpA and band 3 is fOmpA. However, the arrows to the side of the blot suggest that the authors believe band 1 is ufOmpA. How can they be sure? How do the authors know that the indicated band in Figure 5D and Figure 5—figure supplement 1D is mature periplasmic ufOmpA and not precursor OmpA? Without this information it is not possible to make any conclusions about the secretion or assembly of OmpA.

Indeed, there are a few spurious bands in the Western – quite common in antibodies raised in sheep against *E. coli* antigens. However, we were able to eliminate any misidentification and confusion.

Our periplasmic samples were prepared carefully, and were devoid of any contaminating cytosolic proOmpA. We prepared the proOmpA (unfolded of course) and ran this as a new control in a new panel (Figure 5—figure supplement 1C). Clearly proOmpA runs higher than the unfolded OmpA, and was not present in the samples to confuse the analysis.

With controls we show the higher (‘band 1’) and lower (‘band 3’) MW bands represent unfolded and folded OmpA, respectively. The middle one (‘band 2’; Figure 5—figure supplement 1C, red asterisk) only appears in samples prepared from cells grown with arabinose. We do not know what it is, but we show it is present in stationary cells grown in permissive conditions, as well as in a wild type strain. Thus, they should not promote the build-up of unfolded OmpA – so band 2 is not representative of an unfolded species of OmpA, and was ignored.

We clarify these points in the text and include additional data to rule out the interference of proOmpA and ‘band 2’ in the analysis.

2) The mutant SecD protein that lacks the periplasmic domain does not interact with the BAM machinery. Does this mutant behave similarly to the SecD(D519N) mutant in translocation assays?

We show that the variant SecD(D519N) retains its ability to interact with the Bam complex (Figure 5I), but is incapable of promoting efficient OmpA maturation, which we argue is due to loss of PMF driven conformational changes. While SecDF depletion breaks the interaction, with similar consequences. Presumably the P1 depletion will have the same result, also due to loss of the interaction with the Bam complex. We felt that a whole series of new experiments to prove this obvious conclusion would not have been worthwhile.

Words to this effect have been added for further clarity in the Discussion, highlighting the similarity of the effects caused by loss of a functional dynamic interplay, and disconnection of the interaction.

3) The SurA data suggests a function for SecDF in bacteria that lack the BAM machinery, such as Gram-positive bacteria. However, these results are still very preliminary. The author should provide additional experiments demonstrating that the interaction between SecDF and SurA-OmpA is not an interaction artefact. Alternatively, because this interaction is not required for any of the authors' main conclusions, it could be removed from the paper.

Agree. We have removed this from the paper and are in the process of improving and building on this work for another stand-alone paper.

4) More detailed structural characterization would of course be useful in describing the molecular mechanism but the current state convincingly shows the interaction between both translocon complexes. Have the authors tried to further characterize the open and closed states of the HTL complex by cross linking / mass spectrometry (for example before CL addition and glycerol gradient fractionation and after?

Not yet, but this is certainly where we intend to go. We are in the process of trying to reduce the inherent flexibility of the CL^-^stabilised open translocon (not easy), in order to produce a better specimen for high-resolution electron microscopy. This objective is way beyond the scope of this paper. However, as suggested we intend to further characterise the open-state, including its interaction and dynamic interplay with the BAM complex, for a follow up publication.

We have added words to this effect in the Discussion.

Revisions expected in follow-up work:1) SecD was originally identified because mutants have a translocation defect. For example, in one of the papers cited by the authors, Beckwith and colleagues identified the secD locus based on the ability of mutations at this locus to cause defects in translocation. It would therefore be a little surprising if the SecD(D519N) mutant does not cause a translocation defect. If the authors wish to conclude that the SecD(D519N) mutant does not cause a defect in translocation, they should use a more sensitive assay (such as a pulse-chase assay). The assay used by authors measures steady state levels of protein, which could potentially miss transient translocation defects.

We think this has been a misunderstanding, due to terminology. Rather than a translocation defect, it would be better to say a secretion defect – which could mean a problem of translocation through the inner membrane OR beyond. The in vivo experiment we show (Figure 5B) highlights a strong growth defect – consistent with an unspecified secretion defect. It would indeed be difficult to pinpoint this in an in vivo experiment. So we set up an in vitro experiment (Figure 5C), which show the secretion problem is not caused by defective translocation through the Sec machinery at the inner membrane, so it must be something downstream of that.

We have clarified this section of the Results accordingly.